# Modulation of Gut Microbiota through Low-Calorie and Two-Phase Diets in Obese Individuals

**DOI:** 10.3390/nu15081841

**Published:** 2023-04-11

**Authors:** Laurie Lynn Carelli, Patrizia D’Aquila, Francesco De Rango, Armida Incorvaia, Giada Sena, Giuseppe Passarino, Dina Bellizzi

**Affiliations:** 1MEDICAL, Clinical Analysis Laboratory, 87100 Cosenza, Italy; 2Department of Biology, Ecology and Earth Sciences, University of Calabria, 87036 Rende, Italy

**Keywords:** obesity, low-calorie diet, ketogenic diet, gut microbiota, probiotics

## Abstract

Different nutritional regimens have been reported to exert beneficial effects on obesity through the regulation of the composition and function of gut microbiota. In this context, we conducted in obese subjects two dietary interventions consisting of a low-calorie and two-phase (ketogenic plus low-calorie) diet for 8 weeks. Anthropometric and clinical parameters were evaluated at baseline and following the two diets, and gut microbiota composition was assessed by 16S rRNA gene sequencing. A significant reduction was observed for abdominal circumference and insulin levels in the subjects following the two-phase diet. Significant differences in gut microbial composition were observed after treatment compared to the baseline. Both diets induced taxonomic shifts including a decrease in *Proteobacteria*, which are recognized as dysbiosis markers and enrichment of *Verrucomicrobiaceae*, which has recently emerged as an effective probiotic. An increase in *Bacteroidetes*, constituting the so-called good bacteria, was observable only in the two-phase diet. These findings provide evidence that a targeted nutritional regimen and an appropriate use of probiotics can modulate gut microbiota to reach a favorable composition and achieve the balance often compromised by different pathologies and conditions, such as obesity.

## 1. Introduction

The gut microbiota (GM) is a complex microbial community including bacteria, fungi, viruses, and parasites that lives in symbiosis within the human gastrointestinal tract. It performs many important physiological functions that allow the host to achieve intestinal homeostasis [1,2]. In fact, GM is involved in several biological processes such as nutrient extraction, metabolism, immunity as well as biosynthesis of bioactive molecules such as vitamins, folate, riboflavin, biotin, amino acids, and lipids [3,4,5]. Additionally, GM exerts structural and protective functions to strengthen the intestinal epithelium of the host and protect him from pathogens [6]. Gut microbiota varies taxonomically and functionally along the gastrointestinal tract segments and undergoes significant intraindividual variations in composition over infant transition, weaning period, and age [7,8,9]. An increase in the gut community diversity and abundance has been observed throughout life. From three years to adulthood, the predominant phyla were *Firmicutes*, *Bacteroidetes*, and *Actinobacteria*; meanwhile, after the age of 70, a general decrease in *Firmicutes* and an increase in *Bacteroidetes* and *Verrucomicrobia* abundances have been observed [10,11].

Differences in the gut microbiota composition have also been observed among individuals. The origin of this inter-individual plasticity lies in the interplay among GM, dietary and cultural habits, host genetics, and pathological conditions as well as antibiotics use [12,13,14,15]. Many studies are proving that diet represents the main modulator of GM composition in the short and the long term by both directly introducing food-derived microorganisms or promoting or inhibiting the growth of pre-existing ones and indirectly regulating the metabolism or the immune system of the host. More particularly, high-fat, high-sugar, and low-fiber diets reduced *Bacteroidetes*, *Prevotella*, *Lactobacillus* spp., and *Roseburia* spp. Furthermore, a decrease in *Firmicutes*, *Proteobacteria*, *Mollicutes*, *Bacteroides* spp, and *Enterobacteriaceae* was also described in association with a loss of the gut permeability and low-grade systemic inflammation [16,17,18]. The administration of a ketogenic diet induces an abundance of *Bacteroidetes*, *Akkermansia muciniphila*, and *Parabacteroides* spp as well as a decrease in *Firmicutes*, *Proteobacteria*, *Actinobacteria*, *Lactobacillus*, and *Bifidobacterium* in both humans and model organisms [19,20]. The adherence to the Mediterranean diet, which is based on vegetables, high fiber and omega-3 fatty acids and low in saturated fat and animal proteins, displayed high *Prevotella*:*Bacteroides*, *Firmicutes*:*Bacteroidetes*, and *Bifidobacteria*:*Escherichia coli* ratios [21,22]. Finally, a plant-based diet induces an increase in *Bacteroides* and *Prevotella* [23]. Similarly, food components including carbohydrates, fermentable dietary fibers, prebiotics, sweeteners, and emulsifiers as well as vitamins and polyphenols may shape the gut microbiota composition, exerting, in most cases, an increase in the abundance of beneficial microbes [24]. A diet rich in fiber is positively correlated with bacterial richness: the colon microbiota can metabolize complex carbohydrates and oligosaccharides into short-chain fatty acids (SCFAs), which perform a role in the regulation of intestinal pH and induce epigenetic modifications in the host, thus regulating its metabolism [14]. Additionally, the fibers regularize bowel movements and play anti-inflammatory and metabolic effects by reducing Reactive Oxygen Species (ROS) generation and the expression of pro-inflammatory cytokines. Furthermore, they prevent the reabsorption and promote the excretion of bile acids with feces, and they have prebiotic activity, stimulating the growth of beneficial microbes [25,26,27].

A whole series of evidence has also highlighted the relationship between gut microbiota and diseases, including Chronic Inflammatory Disorders, cardiovascular and neurological disorders, and many forms of cancer, diabetes, and obesity [28,29,30,31,32]. More particularly, obesity has been associated with microbial dysbiosis, namely an imbalance in the microbial equilibrium marked by the loss of overall microbial diversity, the increase in the abundance of pathobionts and sulfate-reducing bacteria, and the reduction of health-promoting SCFA-producers [33,34]. The metabolic activity of GM has been linked to the pathogenesis of obesity through the promotion of fat deposition, the increase in intestinal permeability, and chronic low-grade inflammation [35].

Considering the variable diversity and abundance of GM according to dietary habits and the dynamic relationship between obesity and GM, we evaluated the effects of two nutritional interventions in modulating the gut microbiota in a population sample of obese subjects. We analyzed the microbiota composition in individuals subjected to a low-calorie and two-phase (ketogenic plus low-calorie) nutritional regimens for a total of 8 weeks in co-administration with specific foods and probiotic blend.

## 2. Materials and Methods

### 2.1. Study Participants and Diet Plan

After a screening visit by primary care general practitioners, 38 obese volunteers (48 ± 11 years old) from Southern Italy (Calabria) were enrolled in the study. Exclusion criteria were the following: (1) therapy with antibiotics in the last 3 months; (2) use of prebiotics or probiotics in the last 3 months; and (3) history of cancer or suspected inflammatory bowel disease.

One group of individuals (n = 19) was exposed for 8 weeks to a low-calorie diet of 1800 kcal/day with 22.5% proteins, 50% glucids, and 27.5% lipids. The second group (n = 19) was administered a diet plan organized in two phases of 4 weeks each: a first phase based on a ketogenic diet (900 kcal/day) and a second phase based on a low-calorie diet (1200 kcal/day) during which low-in-sugar foods, containing resistant starch, wheat fiber, inulin, lupine protein, modified wheat gluten, and coconut oil (Carbolight and Nutrilight) were used. Furthermore, during the second phase, a transitional nutritional plan was adopted, in which foods such as rice, whole grain bread, fruit, and legumes were gradually introduced to integrate the Carbolight products. In neither of the two diets did we vary the foods to the subjects, but in both, we adopted a balanced nutritional plan that reduced the caloric intake by preserving components that provide adequate amounts of carbohydrates, lipids, proteins, minerals, and vitamins. The daily meal plan of the two diets is schematized in Appendix A.

Both groups were also supplemented with Probactiol HMO COMBI (Metagenics) containing *Lactobacillus acidophilus*, *Bifidobacterium lactis* Bi-07, Vitamin A and D3, 2′-o-fucosyllactose and threonine.

The Ethical Committee of University of Calabria approved the study, and all subjects gave their written informed consent.

### 2.2. Anthropometric and Clinical Measurements

Anthropometric measurements including body weight (kg), height (cm), abdominal circumference (cm), fat mass, and muscle mass were collected at baseline and after diets. Body Mass Index (BMI) was calculated as the ratio between weight and squared height (kg/m^2^). We adopted the classifications in use by the World Health Organization (WHO): underweight—BMI under 18.5 kg/m^2^, normal weight—BMI greater than or equal to 18.5 to 24.9 kg/m^2^, overweight—BMI greater than or equal to 25 to 29.9 kg/m^2^, obesity—BMI greater than or equal to 30 kg/m^2^.

Venous blood sample were drawn using a vacutainer, and clinical measurements such as glycemia, cholesterol, total triglycerides, insulin, cortisol, and glycated hemoglobin were analyzed.

### 2.3. Fecal Sample Collection and Coproculture Analysis

Fecal samples were collected in sterile plastic cups at the beginning of the study (baseline) and after the 8 weeks of the diet plan. They were inoculated onto selective and differential growth media: Salmonella Shigella (SS) agar for the isolation of *Salmonella* spp and some strains of *Shigella* spp, McConkey agar for the detection of *Enterobacteriaceae*, and Man Rogosa Sharpe (MRS) agar for the detection of *Lactobacillus*. To generate an anaerobic environment, a BD GasPak EZ system was used.

### 2.4. Microbial DNA Extraction

Microbial DNA was extracted from feces by a PureLink Microbiome DNA Purification Kit (ThermoFisher Scientifics) according to the manufacturer’s recommendations. Briefly, 0.2 g of feces was resuspended by vortexing in 600 µL of S1-Lysis Buffer and, subsequently, in 100 mL of S2-Lysis Enhancer. Samples were incubated at 65 °C for 10 min, homogenized by bead beating on vortex for 10 min, and centrifuged at 14,000× *g* for 5 min. Then, 400 µL of the supernatants was transferred to a new microcentrifuge tube in the presence of 250 µL of S3-Cleanup Buffer and centrifuged at 14,000× *g* for 2 min, and 500 µL of the isolated supernatants was vortexed in 900 µL of S4-Binding Buffer. Afterwards, 700 µL of samples was loaded onto a spin column-tube and centrifuged at 14,000× *g* for 1 min. Then, 500 µL of S5-Wash Buffer was added to each sample, and columns were centrifuged at 14,000× *g* for 1 min. Microbial DNA samples were eluted by a centrifugation at 14,000× *g* for 1 min in 100 µL of S6-Elution Buffer. The purity and concentration of the DNA obtained were determined through 260/280 nm absorbance measures using the NanoDrop spectrophotometer.

### 2.5. Microbiome Analysis by Next-Generation Sequencing

The variable V3–V4 region of the bacterial 16S rRNA gene (16S ribosomal ribonucleic acid) was sequenced by the company BMR Genomics of Padua through the MiSeq platform (Illumina).

### 2.6. Data Processing and Analysis

The raw data set reads of the full processing of amplicons (fastaq files) were imported using QIIME 2.0 tools version 2021.4.0. Raw reads were pre-processed using Cutadapt. Paired-end reads were demultiplexed and featured tables were constructed by using the Divisive Amplicon Denoising Algorithm (DADA2). Taxonomic assignment was obtained using trained sequences (Operational Taxonomic Units, OTUs at 99%) from the GreenGenes database version 13-8 by the q2-feature-classifier QIIME 2 plugin. To visualize microbiota composition, stacked bar plots were constructed with ggplot2 R-package.

Alpha diversity was assessed using the alpha_rarefaction.py script in QIIME to determine the Shannon index. Alpha diversities were compared using the Wilcoxon paired test. Beta-diversity was calculated in R-vegan package (2.6.0) using the Bray–Curtis index. Statistical significance of beta diversity was determined through the permutational multivariate analysis of variance (PERMANOVA).

To analyze the different OTUs, we used the “edgeR” package in R for the empirical analysis of differential gene expression (DGE). This package uses the relative log expression (RLE) as the default normalization method and assumes a negative binomial distribution model for the counts. The zeros present in count data are modeled using point mass at zero, while remaining log-transformed counts follow a normal distribution.

Statistical analyses were performed using SPSS 20.0 statistical software (SPSS Inc., Chicago, IL, USA). One-way analysis of variance (ANOVA) and Student’s *t*-test were adopted. A *p* value ≤ 0.05 has been considered statistically significant.

## 3. Results

### 3.1. Characteristics of the Study Participants

The anthropometric and clinical characteristics of the study participants at baseline and after the two diets are presented in Table 1 and Appendix A. No significant variations in the anthropometric values were observed after 8 weeks of the low-calorie diet, although there is a tendency for all the parameters to decrease after the diet. A statistically significant decrease in the abdominal circumference was found following the two-phase diet (*p*-value = 0.041). Additionally, a trend toward a reduction was observed for all other parameters.

As for the clinical parameters, in individuals following the low-calorie diet, no changes were observed in the values of glycemia, cholesterol, total triglycerides, insulin, cortisol, and glycated hemoglobin. We found similar results in individuals consuming the two-phase diet with the only exception being that insulin levels decreased significantly after the diet (*p*-value = 0.040; Table 2 and Appendix A).

### 3.2. Coproculture Analysis

Coproculture analysis showed the prevalence in the growth media of *Lactobacilli* and *Bifidobacteria*, although the presence of *Bacteroides* and *Clostridia* has also been detected. It is interesting to observe that in some subjects, *Lactobacilli* appear long and filamentous, while in others, it was more compact. We also noted the detection of *Akkermansia municiphila* in some fecal samples.

### 3.3. OTU Analysis and Microbiota Species Diversity

The total number of Operational Taxonomic Units (OTUs) was equal to 1645 with an average of 104 ± 40 for each sample (ranging from 50 to 188) in the baseline group and of 97 ± 36 (ranging from 48 to 166) in the low-calorie diet. Regarding the two-phase diet, the baseline group was characterized by 134 ± 37 OTUs for each sample (ranging from 80 to 225); meanwhile, the diet group was by 129 ± 47 OTUs (ranging from 54 to 220). As shown in Figure 1, the rarefaction curves of each sample tend to plateau as the sequencing depth increases, demonstrating that the sample sequencing in both low-calorie (Figure 1A) and two-phase (Figure 1B) diets is adequate to capture the entire microbial community, thus guaranteeing the reliability of our research.

The analysis of the relative abundance of OTUs for the two dietary regimens revealed 120 and 122 significantly different OTUs after the low-calorie diet and the two-phase diet (*p* < 0.05), respectively, and bacteria were distributed among five phyla, namely *Actinobacteria*, *Bacteroidetes*, *Firmicutes*, *Proteobacteria*, and *Verrucomicrobia*. Particularly, *Proteobacteria* and *Verrucomicrobia* phyla showed a marked decrease and an increase, respectively, after the low-calorie diet, meanwhile the other phyla have shown greater variability after the two-phase diet. The list and the relative abundance of the significant OTUs after the low-calorie diet and the two-phase diet are reported in Table 3 and Table 4, respectively.

Alpha diversity, which reflects the species diversity of the community, was assessed using the Shannon index. Although the index of the low-calorie diet group was higher and that of the two-phase diet group was lower than that of the baseline groups (Figure 2), the differences were not statistically significant (*p* = 0.83 and *p* = 0.55, respectively).

No significant results in low-calorie (Figure 3A) and two-phase (Figure 3B) diets were also observed for beta diversity assessed using the Bray–Curtis index to evaluate differences in species diversity among samples (*p*-value = 1).

### 3.4. Structure of the Microbiota Associated with Low-Calorie and Two-Phase Diets

The characterization of the gut microbiota of all individuals enrolled in the study at baseline revealed the presence of five bacterial phyla, including *Actinobacteria*, *Bacteroidetes*, *Firmicutes*, *Proteobacteria*, and *Verrucomicrobia*, which *Firmicutes* and *Actinobacteria* represent the predominant ones with a relative abundance of about 79% and 12%, respectively (Figure 4A). Additionally, 15 families, consisting of *Bacteroidaceae*, *Bifidobacteriaceae*, *Clostridiaceae*, *Coriobacteriaceae*, *Enterobacteriaceae*, *Erysipelotrichaceae*, *Lachnospiraceae, Lactobacillaceae*, *Porphyromonadaceae*, *Prevotellaceae*, *Ruminococcaceae*, *Streptococcaceae*, *Turicibacteraceae*, *Veillonellaceae*, and *Verrucomicrobiaceae*, were identified. Among these, *Lachnospiraceae*, *Ruminococcaceae*, and *Bifidobacteriaceae* represented the first three predominant families with a relative abundance of about 50%, 13%, and 10%, respectively (Figure 4B).

Changes in bacterial abundance of more than 1.5-fold ratio induced by the diet with respect to the baseline were also considered relevant. We found that the low-calorie diet induces, with respect to the baseline, an enrichment, at Phylum level, in *Verrucomicrobia* (3.9-fold) and a decrease in *Proteobacteria* (6.1-fold) (Figure 4A). At the level of Family, we observed that the diet induces, with respect to the baseline, enrichment in *Lactobacillaceae* (1.5-fold), *Turicibacteraceae* (1.8-fold), and *Verrucomicrobiaceae* (3.9-fold), and a reduction in *Enterobacteriaceae* (6.4-fold), and *Prevotellaceae* (2.8-fold) (Figure 4B).

The two-phase diet induces, with respect to the baseline, enrichment in *Bacteroidetes* (2.1-fold) and in *Verrucomicrobia* (5.5-fold) Phyla and a decrease in *Proteobacteria* (3.1-fold) Phylum (Figure 4A). An increase in the abundance of *Porphyromonadaceae* (2.4-fold), *Veillonellaceae* (3.5-fold), and *Verrucomicrobiaceae* (5.4-fold) as well as a decrease in the abundance of *Enterobacteriaceae* (4.2-fold), *Streptococcaceae* (1.9-fold), and *Turicibacteraceae* (2.2-fold) families was also observed (Figure 4B).

## 4. Discussion

The diet greatly influences the composition, diversity, and functional activity of gut microbiota, significantly affecting human health [15]. Different factors determine perturbations that induce the onset of dysbiosis phenomena, which is characterized mainly by a lowering of microbial diversity and an alteration in the symbiotic relationship with the guest [36]. Dysbiosis in the microbiota appears strongly connected to numerous chronic pathologies from metabolic, inflammatory, neurological, cardiovascular, and respiratory disorders [37]. Since the well-being of the intestinal microbiota generally reflects that of its host, numerous therapeutic interventions are aimed at improving dysbiosis conditions and, therefore, pathological conditions, including the use of probiotics. Obesity is a complex, multifactorial disease due to various factors including the host genetic background, decreased physical activity, and excess food intake [38,39]. A series of microbiota markers associated with this pathology have been identified. Recently, research efforts have focused on identifying bacterial taxa involved in the development of obesity.

In this study, we report changes in the composition of fecal bacteria of obese individuals fed with two different dietary regimens: an 8-week low-calorie diet and a two-phase diet in which the initial phase of 4 weeks consisted of a ketogenic diet and the second 4 weeks consisted of a low-calorie diet. Furthermore, in the two-phase dietary intervention, low in sugar, source of protein, and rich in fiber foods were given along with a multivitamin to make up for the lack of fruit consumption in the ketogenic diet. A probiotic containing *Lactobacillus* and *Bifidobacteria* was also administered during the two dietary regimens.

The low-calorie diet is commonly considered optimal for managing obesity for its versatility and flexibility and may be helpful in restoring the gut microbiome dysbiosis in obese patients.

The two-phase diet we adopted combines the well-known benefits of a ketogenic diet on weight loss with the previously described advantages of the low-calorie-diet to prevent the outbreak of some negative effects correlated to a long-time ketogenic diet, such as increased risk of kidney stones, hypoproteinemia, and osteoporosis, and increased blood levels of uric acid [40]. Two-phase dietary approaches have already been described in the literature, although differences in terms of duration of each phase, caloric intake and macro- and micro-nutritional supplementation make their generalization difficult [41,42,43]. In this context, we also opted for the administration of Carbolight Products from the LightFlow Company, which are poor in carbohydrates and relatively rich in proteins of vegetable origin, to enrich the nutritional regimen with fibers [44]. It is interesting to note that the two-phase nutritional approach we adopted, compared with the low-calorie diet, was more effective in inducing a decrease in abdominal circumference and in insulin levels.

The gut microbiota composition of healthy non-obese individuals consists, in order of relative abundance, of *Bacteroidetes* (73%), *Firmicutes* (22%), *Proteobacteria* (2%) and *Actinobacteria* (1.8%) [45]. In our study, we found that the obese subject constituting the baseline group exhibited an increased abundance of *Firmicutes* (70%) at the expense of *Bacteroidetes* (4%), further reinforcing evidence already reported in the literature that consider the high ratio of *Firmicutes*: *Bacteroidetes* as a hallmark of obesity [39,46,47]. It has been proposed that *Firmicutes* take out energy from foods more effectively than *Bacteroidetes*, thus supporting the efficient absorption of calories with subsequent weight gain. In line with Turnbaugh et al., besides the above two phyla, we observed in the same persons high levels of *Actinobacteria* and *Proteobacteria* [48]. Furthermore, as stated by Clarke et al., obese participants in our study contained a lower proportion of *Verrucomicrobia* [49].

In addition, the abundance of gut microbiota in individuals subjected to the two nutritional regimens was different from that of the same individuals before starting the diet.

Interestingly, after the two nutritional regimens, the amount of *Firmicutes* remains globally unchanged, although we observed variations relating to specific families, according to the type of diet that are unrepresentative in terms of the percentage of the entire community. Still, an increase in *Bacteroidetes* occurred after the two-phase diet, which could be explained considering the high uptake of proteins in the first phase followed by an increase administration in soluble fiber intake in the second. Therefore, it is plausible to hypothesize that the two-phase diet is more efficient than the low-calorie diet in promoting the abundance of members of *Bacteroidetes*, so-called good bacteria because they produce favorable metabolites, such as SCFAs. The data obtained in our study demonstrated that both nutritional regimens decrease the abundance of *Proteobacteria* but do not affect that of *Actinobacteria*. This result seems very interesting, since an increased prevalence of *Proteobacteria* in the gut microbiome is a potential diagnostic signature of dysbiosis and risk of disease [50]. *Proteobacteria* is the phylum most conditioned by the Western diet rich in fats, sugars, and animal proteins, and, simultaneously, it is more linked to the metabolic and inflammatory states of the host [51]. Indeed, in obese subjects, the gut-derived endotoxin lipopolysaccharide (LPS), of which *Proteobacteria* is a major source, binds to the TLR-CD14-MD-2 complex and activates the Toll-like receptor 4 (TLR4) signaling, resulting in the activation of the expression of IFN inducible genes and pro-inflammatory mediators [52]. What is more, Alexander et al. demonstrated the beneficial effect of the natural fiber inulin in decreasing the abundance of *Proteobacteria* and in increasing the abundance of some *Firmicutes* [53]. The decrease in the relative abundance of *Proteobacteria* we observed following the two diets suggests that the reduced fat uptake associated with the consumption of fiber and probiotics could reduce and/or eliminate the chronic inflammatory state of the body. Furthermore, resistance to commonly used antibiotics, a problem that has been assuming enormous importance from some years, characterizes many members of this phylum. Therefore, the decrease in *Proteobacteria*, more specifically of the *Enterobacteriaceae*, which is highlighted as the adoption of a probiotic in association with nutritional regimens of only 8 weeks, can be considered a starting point for the eradication of many infections and their complications.

Since the health benefits exerted by the administration of probiotics in human health have been extensively described, both nutritional treatments were supplemented by *Lactobacillus acidophilus* and *Bifidobacterium lactis* Bi-07, representing the most studied bacterial species recommended for dietary use [54]. In vivo studies carried out in different mice models revealed that the administration of these probiotics induces an improvement in insulin sensitivity and lipid profile with the decreased level of total cholesterol, LDL cholesterol, and plasma TG, the reduction of pro-inflammatory genes including IL-6, tumor necrosis factor-a, IL-1b, and IL-17, and the increase in IL-10 [55]. The supplementation of overweight and obese adults with *Lactobacilli* and *Bifidobacteria* significantly reduced body weight, BMI, abdominal circumference, and waist-to-height ratio in a free-living overweight/obese population and improves well-being [56]. It has been reported that some *Bifidobacterium* spp. and *Lactobacillus* spp. promote the synthesis of conjugated linoleic acid (CLA), which has been shown to modulate body weight by reducing energy intake and improving metabolic rate and lipolysis [57]. Additionally, the administration of the prebiotic 2′-fucosyllactose, the most prevalent human milk oligosaccharide (HMO) present in human breast milk, has been demonstrated to counteract gut permeability and insulin resistance, improve lipid utilization, and decrease de novo lipogenesis, thus reducing the obesity-associated steatosis [58]. It follows that the combined use of pro- and prebiotics, which are directly involved in the reduction of the state of chronic systemic inflammation and in the promotion of lipolysis as well as associated with the two nutritional approaches used in this study for the treatment of obesity, appears particularly effective not just for weight loss but for a global restoration of systemic well-being, acting through the improvement of gut microbiota.

Despite the oral consumption of probiotics, we did not find significant abundance changes for *Bifidobacteriaceae*, thus confirming similar evidence reported in the literature [59]. An increase in *Lactobacillales* was observed after the sole low-calorie diet. However, the presence of these bacteria is evident in the coprocultures we carried out.

Furthermore, the significant increase in *Verrucomicrobiaceae*, more specifically in *Akkermansia muciniphila*, a mucin-degrading bacterium, in subjects administered with both nutritional regimens is of particular interest. It seems to play a key role in metabolic and gastrointestinal pathologies by mainly improving the functionality of the intestinal barrier [60]. Note that due to its highly promising activities against obesity and diabetes, *Akkermansia* has drawn intensive interest for research so much that it was recently marketed as a probiotic. Similarly, after the administration of the two-phase diet, the significant increase we observed in the genus *Roseburia*, a butyrate-producing bacteria, appears of relevance. Some evidence reported that butyrate, by the activation of AMPK, the increasing ATP consumption, and the induction of PGC-1α activity, promotes mitochondrial function and the expression of genes involved in lipolysis and fatty acid β-oxidation [61]. Therefore, the rise in the *Roseburia* genus seems directly involved in the increase in fat mobilization and the promotion of energy expenditure, suggesting that the assumption of this dietary regimen represents an effective strategy for the control and treatment of obesity. Additionally, the rise of *Roseburia* abundance has been found to exert protective effects on the development of type 2 diabetes by increasing insulin sensitivity, as well as against all inflammatory pathologies, by inhibiting the synthesis of proinflammatory cytokines and the balance of the immune system [62].

## 5. Conclusions

The results we obtained demonstrate that the adoption of specific nutritional interventions associated with the administration of effective probiotics, in just 8 weeks, may modify the structure of the gut microbiota, affecting bacteria whose functions have been demonstrated to be correlated with the health status in humans. Particularly, the increase in the phylum *Bacteroidetes*, with a shift of the ratio *Firmicutes*:*Bacteroidetes* toward values closer to that found in non-obesity conditions, associated with the increase in genera *Akkermansia* and *Roseburia*, let us consider the two-phase nutritional approach as the most effective in restoring the balance at the level of the gut microbial community in obesity. It follows that an adequate combination of nutritional intake and probiotics may modify the intestinal microbiota by enhancing those species, genera, and families, which is useful to contrast the dysbiosis and weaken the state of chronic systemic inflammation that characterize different systemic pathologies.

## Figures and Tables

**Figure 1 nutrients-15-01841-f001:**
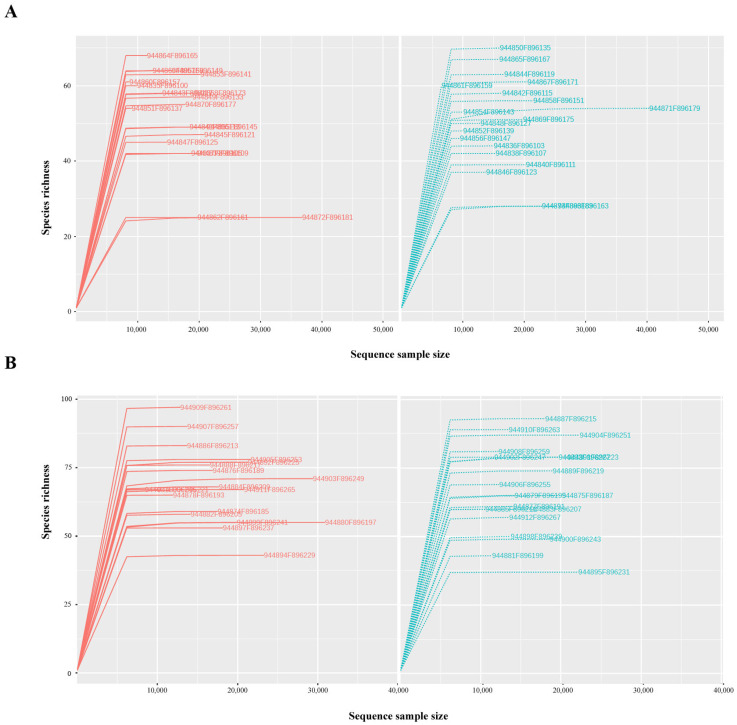
Rarefaction curves before (red) and after (blue) low-calorie (**A**) and two-phase (**B**) diet. Every curve corresponds to a single sample.

**Figure 2 nutrients-15-01841-f002:**
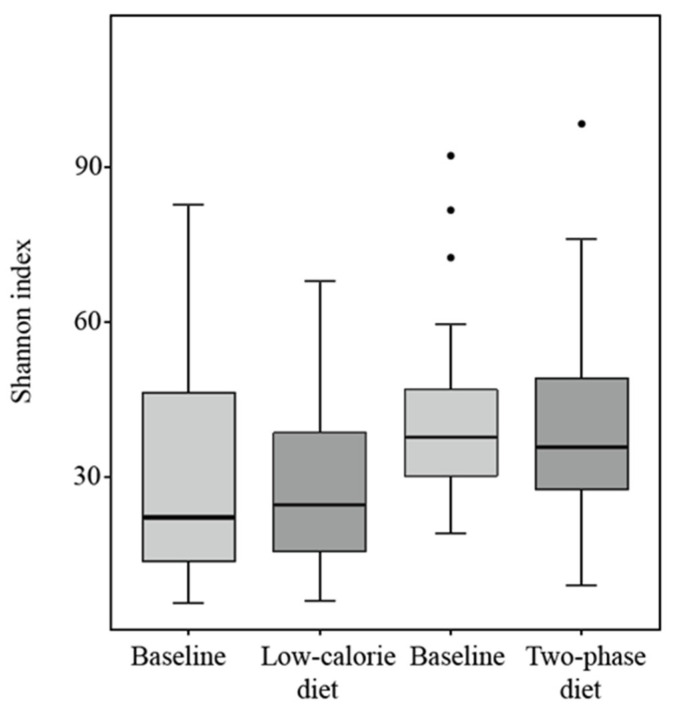
Alpha diversity evaluated by Shannon index at baseline and after the low-calorie and two-phase diets.

**Figure 3 nutrients-15-01841-f003:**
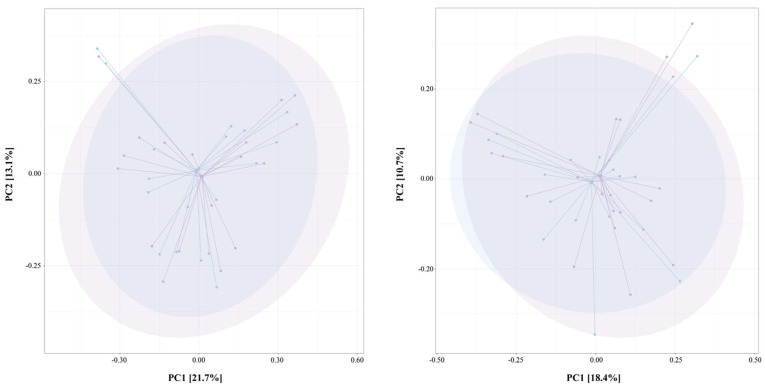
Principal Coordinate Analysis of Bray–Curtis distance for beta-diversity evaluation between baseline (blue samples) and low-calorie (on the **left**) and two-phase (on the **right**) diets (violet sample). PC1 and PC2 represent the top two principal coordinates that captured most of the diversity.

**Figure 4 nutrients-15-01841-f004:**
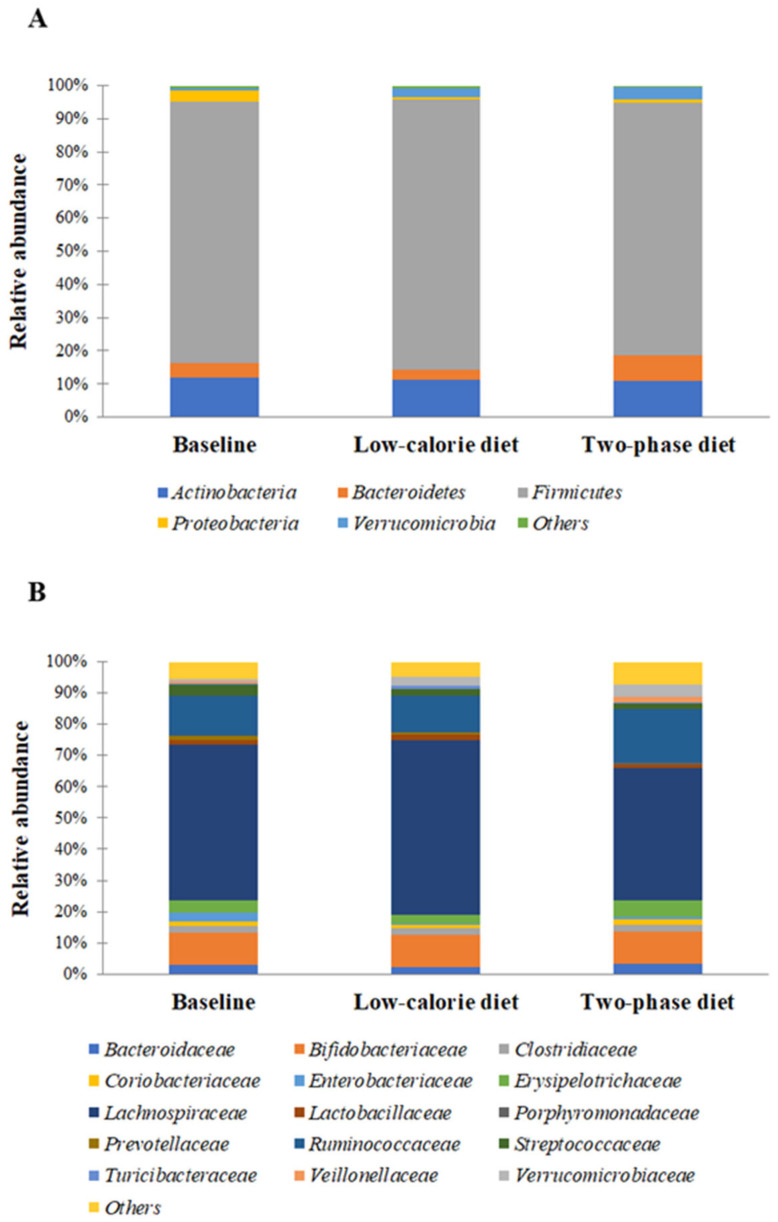
Structure of the gut microbiota at baseline and after low-calorie and two-phase diets. Relative abundance of Phyla (**A**) and Families (**B**) is reported.

**Table 1 nutrients-15-01841-t001:** Mean values of anthropometric measurements of the total number of participants (N) belonging to baseline, low-calorie diet, and two-phase diet groups. SD: Standard Deviation.

Group	Weight (kg)		BMI		Abdominal Circumference		Fat Mass		Muscular Mass	
	Baseline	Low-Calorie Diet	Baseline	Low-Calorie Diet	Baseline	Low-Calorie Diet	Baseline	Low-Calorie Diet	Baseline	Low-Calorie Diet
N	19	19	19	19	19	19	19	19	19	19
Mean	91	86.421	34.46	32.708	112.737	108.053	34.437	31.447	39.853	38.642
SD	20.79	20.797	6.637	6.63	13.457	13.966	16.054	16.415	8.29	8.03
*p*	0.307		0.273		0.156		0.307		0.511	
Group	Weight (kg)		BMI		Abdominal circumference		Fat mass		Muscular mass	
	Baseline	Two-phase diet	Baseline	Two-phase diet	Baseline	Two-phase diet	Baseline	Two-phase diet	Baseline	Two-phase diet
N	19	19	19	19	19	19	19	19	19	19
Mean	93.121	86.242	33.695	31.212	113.474	105.789	32.774	27.926	42.963	42.274
SD	17.038	16.03	5.674	5.425	10.777	9.09	10.28	10.427	9.314	9.225
*p*	0.130		0.215		0.041		0.140		0.540	

**Table 2 nutrients-15-01841-t002:** Mean values of clinical measurements of the total number of participants (N) belonging to baseline, low-calorie diet, and two-phase diet groups. SD: Standard Deviation.

	Glucose		Cholesterol		Triglyceride		Insulin		Cortisol		Glycated Hemoglobin	
Group	Baseline	Low-Calorie Diet	Baseline	Low-Calorie Diet	Baseline	Low-Calorie Diet	Baseline	Low-Calorie Diet	Baseline	Low-Calorie Diet	Baseline	Low-Calorie Diet
N	19	19	19	19	19	19	19	19	19	19	19	19
Mean	100.789	95	186.053	189.211	116.368	104.053	14.479	13.722	14.158	13.932	5.811	5.753
SD	18.546	17.346	34.725	36.163	48.419	40.923	14.326	22.925	6.946	3.816	0.716	0.619
*p*	0.188		0.953		0.530		0.116		0.502		0.930	
	Glucose		Cholesterol		Triglyceride		Insulin		Cortisol		Glycated hemoglobin	
Group	Baseline	Two-phase diet	Baseline	Two-phase diet	Baseline	Two-phase diet	Baseline	Two-phase diet	Baseline	Two-phase diet	Baseline	Two-phase diet
N	19	19	19	19	19	19	19	19	19	19	19	19
Mean	94.526	92.842	203.368	198.684	116.263	105.947	14.832	12.007	14.747	13.868	5.489	5.626
SD	9.252	10.777	34.587	38.662	43.735	48.555	8.744	13.082	3.897	4.828	0.501	0.484
*p*	0.895		0.826		0.090		0.040		0.838		0.319	

**Table 3 nutrients-15-01841-t003:** Changes in the Operational Taxonomic Units (OTUs) induced by the low-calorie diet with respect to the baseline and the relative taxonomic classification. log2FC (fold change). FDR: False Discovery Rate.

OTU	log2FC	*p*-Value	FDR	Kingdom	Phylum	Class	Order	Family	Genus	Species
1	5.6542	1.7965 × 10^−8^	9.7909 × 10^−6^	*Bacteria*	*Firmicutes*	*Clostridia*	*Clostridiales*	*Ruminococcaceae*	*Ruminococcus*	*bromii*
2	−4.1949	1.6834 × 10^−6^	4.5872 × 10^−4^	*Bacteria*	*Firmicutes*	*Clostridia*	*Clostridiales*	*Ruminococcaceae*	*Ruminococcus*	
3	−3.8608	4.4462 × 10^−6^	7.8653 × 10^−4^	*Bacteria*	*Bacteroidetes*	*Bacteroidia*	*Bacteroidales*	*Bacteroidaceae*	*Bacteroides*	
4	−3.6553	5.7727 × 10^−6^	7.8653 × 10^−4^	*Bacteria*	*Firmicutes*	*Clostridia*	*Clostridiales*	*Ruminococcaceae*		
5	4.9451	1.0752 × 10^−5^	8.6544 × 10^−4^	*Bacteria*	*Bacteroidetes*	*Bacteroidia*	*Bacteroidales*	*Bacteroidaceae*	*Bacteroides*	
6	−3.5161	1.2704 × 10^−5^	8.6544 × 10^−4^	*Bacteria*	*Bacteroidetes*	*Bacteroidia*	*Bacteroidales*	*Bacteroidaceae*	*Bacteroides*	*uniformis*
7	−3.9301	1.7322 × 10^−5^	1.0490 × 10^−3^	*Bacteria*	*Bacteroidetes*	*Bacteroidia*	*Bacteroidales*	*Bacteroidaceae*	*Bacteroides*	
8	−3027	2.9839 × 10^−5^	1.4784 × 10^−3^	*Bacteria*	*Bacteroidetes*	*Bacteroidia*	*Bacteroidales*	*Porphyromonadaceae*	*Parabacteroides*	*distasonis*
9	−3.3407	5.2814 × 10^−5^	2.3986 × 10^−3^	*Bacteria*	*Firmicutes*	*Clostridia*	*Clostridiales*	*Lachnospiraceae*	*Lachnobacterium*	
10	4.9124	5.9602 × 10^−5^	2.4987 × 10^−3^	*Bacteria*	*Firmicutes*	*Clostridia*	*Clostridiales*	*Lachnospiraceae*	*Blautia*	
11	3.3116	7.8375 × 10^−5^	2.8684 × 10^−3^	*Bacteria*	*Bacteroidetes*	*Bacteroidia*	*Bacteroidales*	*Bacteroidaceae*	*Bacteroides*	*ovatus*
12	−2.7334	7.8948 × 10^−5^	2.8684 × 10^−3^	*Bacteria*	*Bacteroidetes*	*Bacteroidia*	*Bacteroidales*	*Bacteroidaceae*	*Bacteroides*	
13	−2.6188	1.0949 × 10^−4^	3.7297 × 10^−3^	*Bacteria*	*Proteobacteria*	*Alphaproteobacteria*	*RF32*			
14	2.4015	1.3789 × 10^−4^	4.2072 × 10^−3^	*Bacteria*	*Firmicutes*	*Erysipelotrichi*	*Erysipelotrichales*	*Erysipelotrichaceae*		
15	2.3058	1.3895 × 10^−4^	4.2072 × 10^−3^	*Bacteria*	*Firmicutes*	*Clostridia*	*Clostridiales*	*Ruminococcaceae*		
16	−2.5623	1.5965 × 10^−4^	4.4846 × 10^−3^	*Bacteria*	*Firmicutes*	*Clostridia*	*Clostridiales*	*Ruminococcaceae*	*Ruminococcus*	
17	−2.8927	1.6457 × 10^−4^	4.4846 × 10^−3^	*Bacteria*	*Firmicutes*	*Clostridia*	*Clostridiales*	*Ruminococcaceae*	*Faecalibacterium*	*prausnitzii*
18	−2.4442	1.9061 × 10^−4^	4.9467 × 10^−3^	*Bacteria*	*Firmicutes*	*Clostridia*	*Clostridiales*	*Lachnospiraceae*	*Ruminococcus*	*gnavus*
19	−2.3237	2.0365 × 10^−4^	5.0449 × 10^−3^	*Bacteria*	*Firmicutes*	*Clostridia*	*Clostridiales*	*Ruminococcaceae*	*Ruminococcus*	
20	2.5439	2.4671 × 10^−4^	5.8458 × 10^−3^	*Bacteria*	*Verrucomicrobia*	*Verrucomicrobiae*	*Verrucomicrobiales*	*Verrucomicrobiaceae*	*Akkermansia*	*muciniphila*
21	−2.2128	2.7378 × 10^−4^	6.0787 × 10^−3^	*Bacteria*	*Firmicutes*	*Clostridia*	*Clostridiales*	*Ruminococcaceae*	*Oscillospira*	
22	2.2752	2.7884 × 10^−4^	6.0787 × 10^−3^	*Bacteria*	*Firmicutes*	*Clostridia*	*Clostridiales*	*Ruminococcaceae*	*Ruminococcus*	*callidus*
23	2.8103	3.0214 × 10^−4^	6.3333 × 10^−3^	*Bacteria*	*Firmicutes*	*Clostridia*	*Clostridiales*	*Lachnospiraceae*	*Roseburia*	
24	−2.2536	3.4564 × 10^−4^	6.8203 × 10^−3^	*Bacteria*	*Firmicutes*	*Clostridia*	*Clostridiales*	*Clostridiaceae*	*Clostridium*	*paraputrificum*
25	−2.2567	3.5040 × 10^−4^	6.8203 × 10^−3^	*Bacteria*	*Firmicutes*	*Clostridia*	*Clostridiales*	*Lachnospiraceae*	*Ruminococcus*	*gnavus*
26	−2.5453	3.9285 × 10^−4^	7.3829 × 10^−3^	*Bacteria*	*Firmicutes*	*Clostridia*	*Clostridiales*			
27	−2.6037	5.1619 × 10^−4^	9.3775 × 10^−3^	*Bacteria*	*Firmicutes*	*Clostridia*	*Clostridiales*	*Ruminococcaceae*		
28	3.6789	7.6258 × 10^−4^	1.3407 × 10^−2^	*Unassigned*						
29	−1.7461	7.9325 × 10^−4^	1.3510 × 10^−2^	*Bacteria*	*Firmicutes*	*Clostridia*	*Clostridiales*	*Ruminococcaceae*	*Oscillospira*	
30	−2.3435	9.0130 × 10^−4^	1.4885 × 10^−2^	*Bacteria*	*Bacteroidetes*	*Bacteroidia*	*Bacteroidales*	*Bacteroidaceae*	*Bacteroides*	*caccae*
31	1.8158	1.2923 × 10^−3^	2.0715 × 10^−2^	*Bacteria*	*Firmicutes*	*Clostridia*	*Clostridiales*	*Lachnospiraceae*	*Blautia*	*producta*
32	−3.7721	1.3925 × 10^−3^	2.1545 × 10^−2^	*Bacteria*	*Proteobacteria*	*Gammaproteobacteria*	*Enterobacteriales*	*Enterobacteriaceae*	*Escherichia*	*coli*
33	2.8375	1.4231 × 10^−3^	2.1545 × 10^−2^	*Bacteria*	*Bacteroidetes*	*Bacteroidia*	*Bacteroidales*	*Bacteroidaceae*	*Bacteroides*	
34	1.5655	1.5023 × 10^−3^	2.2129 × 10^−2^	*Bacteria*	*Firmicutes*	*Clostridia*	*Clostridiales*	*Mogibacteriaceae*		
35	2.0469	1.7060 × 10^−3^	2.4426 × 10^−2^	*Bacteria*	*Firmicutes*	*Clostridia*	*Clostridiales*	*Ruminococcaceae*	*Ruminococcus*	
36	−2.76	1.7479 × 10^−3^	2.4426 × 10^−2^	*Bacteria*	*Bacteroidetes*	*Bacteroidia*	*Bacteroidales*	*Bacteroidaceae*	*Bacteroides*	
37	2.1734	1.8406 × 10^−3^	2.5079 × 10^−2^	*Bacteria*	*Actinobacteria*	*Coriobacteriia*	*Coriobacteriales*	*Coriobacteriaceae*	*Adlercreutzia*	
38	1728	1.8913 × 10^−3^	2.5140 × 10^−2^	*Bacteria*	*Bacteroidetes*	*Bacteroidia*	*Bacteroidales*	*Bacteroidaceae*	*Bacteroides*	
39	2.5167	2.2825 × 10^−3^	2.9618 × 10^−2^	*Bacteria*	*Firmicutes*	*Clostridia*	*Clostridiales*	*Ruminococcaceae*	*Gemmiger*	*formicilis*
40	1537	2.3745 × 10^−3^	3.0096 × 10^−2^	*Bacteria*	*Firmicutes*	*Clostridia*	*Clostridiales*			
41	−1.6222	2.5844 × 10^−3^	3.2011 × 10^−2^	*Bacteria*	*Firmicutes*	*Bacilli*	*Lactobacillales*			
42	3.3422	2.8103 × 10^−3^	3.3346 × 10^−2^	*Bacteria*	*Verrucomicrobia*	*Verrucomicrobiae*	*Verrucomicrobiales*	*Verrucomicrobiaceae*	*Akkermansia*	*muciniphila*
43	−1.5857	2.8145 × 10^−3^	3.3346 × 10^−2^	*Bacteria*	*Bacteroidetes*	*Bacteroidia*	*Bacteroidales*	*Odoribacteraceae*	*Odoribacter*	
44	−2.3646	3.1709 × 10^−3^	3.6769 × 10^−2^	*Bacteria*	*Firmicutes*	*Clostridia*	*Clostridiales*	*Lachnospiraceae*	*Coprococcus*	*eutactus*
45	2.2174	3.7102 × 10^−3^	4.1751 × 10^−2^	*Bacteria*	*Bacteroidetes*	*Bacteroidia*	*Bacteroidales*	*Bacteroidaceae*	*Bacteroides*	*caccae*
46	−3.0265	3.7538 × 10^−3^	4.1751 × 10^−2^	*Bacteria*	*Firmicutes*	*Clostridia*	*Clostridiales*	*Lachnospiraceae*	*Blautia*	
47	−1.6015	3.9757 × 10^−3^	4.2879 × 10^−2^	*Bacteria*	*Bacteroidetes*	*Bacteroidia*	*Bacteroidales*	*Rikenellaceae*		
48	−1.8671	4.0126 × 10^−3^	4.2879 × 10^−2^	*Bacteria*	*Bacteroidetes*	*Bacteroidia*	*Bacteroidales*	*Porphyromonadaceae*	*Parabacteroides*	*distasonis*
49	−2.6589	4.1354 × 10^−3^	4.3342 × 10^−2^	*Bacteria*	*Bacteroidetes*	*Bacteroidia*	*Bacteroidales*	*Bacteroidaceae*	*Bacteroides*	
50	−2.3605	4.8328 × 10^−3^	4.8729 × 10^−2^	*Bacteria*	*Bacteroidetes*	*Bacteroidia*	*Bacteroidales*	*Rikenellaceae*		
51	−3.4034	4.8793 × 10^−3^	4.8729 × 10^−2^	*Bacteria*	*Firmicutes*	*Clostridia*	*Clostridiales*	*Ruminococcaceae*	*Faecalibacterium*	*prausnitzii*
52	1622	4.9176 × 10^−3^	4.8729 × 10^−2^	*Bacteria*	*Bacteroidetes*	*Bacteroidia*	*Bacteroidales*	*Barnesiellaceae*		
53	1.7774	5.0856 × 10^−3^	4.9494 × 10^−2^	*Bacteria*	*Firmicutes*	*Clostridia*	*Clostridiales*	*Lachnospiraceae*		
54	2.0009	5.4657 × 10^−3^	5.1412 × 10^−2^	*Bacteria*	*Bacteroidetes*	*Bacteroidia*	*Bacteroidales*	*Bacteroidaceae*	*Bacteroides*	
55	−2.2202	5.4714 × 10^−3^	5.1412 × 10^−2^	*Bacteria*	*Bacteroidetes*	*Bacteroidia*	*Bacteroidales*	*Paraprevotellaceae*	*Prevotella*	
56	1.5872	7.4790 × 10^−3^	6.9086 × 10^−2^	*Bacteria*	*Firmicutes*	*Clostridia*	*Clostridiales*			
57	1.5462	7.9992 × 10^−3^	7.2033 × 10^−2^	*Bacteria*	*Firmicutes*	*Clostridia*	*Clostridiales*	*Lachnospiraceae*	*Dorea*	
58	2554	8.2140 × 10^−3^	7.2033 × 10^−2^	*Bacteria*	*Firmicutes*	*Clostridia*	*Clostridiales*	*Lachnospiraceae*	*Ruminococcus*	*gnavus*
59	−1.9733	8.2894 × 10^−3^	7.2033 × 10^−2^	*Bacteria*	*Bacteroidetes*	*Bacteroidia*	*Bacteroidales*	*Bacteroidaceae*	*Bacteroides*	*ovatus*
60	1.4748	8.3267 × 10^−3^	7.2033 × 10^−2^	*Bacteria*	*Firmicutes*	*Clostridia*	*Clostridiales*	*Lachnospiraceae*	*Anaerostipes*	
61	−2.4647	8.8774 × 10^−3^	7.5596 × 10^−2^	*Bacteria*	*Bacteroidetes*	*Bacteroidia*	*Bacteroidales*	*Bacteroidaceae*	*Bacteroides*	
62	1295	9.7992 × 10^−3^	8.2163 × 10^−2^	*Bacteria*	*Firmicutes*	*Clostridia*	*Clostridiales*	*Lachnospiraceae*	*Lachnospira*	
63	2.1263	1.0165 × 10^−2^	8.2829 × 10^−2^	*Bacteria*	*Firmicutes*	*Clostridia*	*Clostridiales*	*Lachnospiraceae*	*Dorea*	
64	−2.0481	1.0183 × 10^−2^	8.2829 × 10^−2^	*Bacteria*	*Bacteroidetes*	*Bacteroidia*	*Bacteroidales*	*Bacteroidaceae*	*Bacteroides*	
65	1.5517	1.0551 × 10^−2^	8.4560 × 10^−2^	*Bacteria*	*Firmicutes*	*Clostridia*	*Clostridiales*	*Ruminococcaceae*	*Oscillospira*	
66	−3.0346	1.1240 × 10^−2^	8.6544 × 10^−4^	*Bacteria*	*Firmicutes*	*Clostridia*	*Clostridiales*	*Lachnospiraceae*	*Blautia*	
67	−4.4056	1.1350 × 10^−2^	8.6544 × 10^−4^	*Bacteria*	*Firmicutes*	*Clostridia*	*Clostridiales*			
68	−1.7762	1.1431 × 10^−2^	9.0288 × 10^−2^	*Bacteria*	*Bacteroidetes*	*Bacteroidia*	*Bacteroidales*	*Bacteroidaceae*	*Bacteroides*	*plebeius*
69	1205	1.2867 × 10^−2^	1.0018 × 10^−1^	*Bacteria*	*Firmicutes*	*Clostridia*	*Clostridiales*	*Ruminococcaceae*	*Ruminococcus*	
70	1882	1.3794 × 10^−2^	1.0588 × 10^−1^	*Bacteria*	*Firmicutes*	*Clostridia*	*Clostridiales*	*Ruminococcaceae*		
71	2.2378	1.4732 × 10^−2^	1.1024 × 10^−1^	*Bacteria*	*Bacteroidetes*	*Bacteroidia*	*Bacteroidales*	*Bacteroidaceae*	*Bacteroides*	
72	−2.0184	1.4766 × 10^−2^	1.1024 × 10^−1^	*Bacteria*	*Bacteroidetes*	*Bacteroidia*	*Bacteroidales*	*Bacteroidaceae*	*Bacteroides*	
73	1.1121	1.5145 × 10^−2^	1.1131 × 10^−1^	*Bacteria*	*Firmicutes*	*Bacilli*	*Lactobacillales*			
74	1.7359	1.5317 × 10^−2^	1.1131 × 10^−1^	*Bacteria*	*Bacteroidetes*	*Bacteroidia*	*Bacteroidales*	*Bacteroidaceae*	*Bacteroides*	*ovatus*
75	1.1449	1.5660 × 10^−2^	1.1230 × 10^−1^	*Bacteria*	*Firmicutes*	*Clostridia*	*Clostridiales*	*Lachnospiraceae*	*Blautia*	
76	−1.2542	1.5938 × 10^−2^	1.1281 × 10^−1^	*Bacteria*	*Firmicutes*	*Bacilli*	*Gemellales*	*Gemellaceae*		
77	−2.2446	1.7281 × 10^−2^	1.1954 × 10^−1^	*Bacteria*	*Bacteroidetes*	*Bacteroidia*	*Bacteroidales*	*Bacteroidaceae*	*Bacteroides*	*uniformis*
78	−1.4053	1.7327 × 10^−2^	1.1954 × 10^−1^	*Bacteria*	*Firmicutes*	*Clostridia*	*Clostridiales*	*Ruminococcaceae*	*Oscillospira*	
79	−2.5692	1.7952 × 10^−2^	1.2142 × 10^−1^	*Bacteria*	*Proteobacteria*	*Gammaproteobacteria*	*Enterobacteriales*	*Enterobacteriaceae*		
80	−1.8267	1.8047 × 10^−2^	1.2142 × 10^−1^	*Bacteria*	*Bacteroidetes*	*Bacteroidia*	*Bacteroidales*	*Bacteroidaceae*	*Bacteroides*	
81	1.6559	1.8470 × 10^−2^	1.2276 × 10^−1^	*Bacteria*	*Firmicutes*	*Clostridia*	*Clostridiales*	*Ruminococcaceae*	*Oscillospira*	
82	2.1136	1.8825 × 10^−2^	1.2361 × 10^−1^	*Bacteria*	*Firmicutes*	*Bacilli*	*Lactobacillales*	*Streptococcaceae*	*Streptococcus*	
83	1.6886	1.9076 × 10^−2^	1.2377 × 10^−1^	*Bacteria*	*Firmicutes*	*Clostridia*	*Clostridiales*	*Lachnospiraceae*	*Blautia*	*producta*
84	−2.9896	1.9290 × 10^−2^	1.0513 × 10^−3^	*Bacteria*	*Bacteroidetes*	*Bacteroidia*	*Bacteroidales*	*Bacteroidaceae*	*Bacteroides*	*uniformis*
85	−2.0809	1.9874 × 10^−2^	1.2675 × 10^−1^	*Bacteria*	*Firmicutes*	*Clostridia*	*Clostridiales*	*Ruminococcaceae*	*Butyricicoccus*	*pullicaecorum*
86	−1.2734	2.0136 × 10^−2^	1.2675 × 10^−1^	*Bacteria*	*Bacteroidetes*	*Bacteroidia*	*Bacteroidales*	*Bacteroidaceae*	*Bacteroides*	
87	−1.7441	2.0233 × 10^−2^	1.2675 × 10^−1^	*Bacteria*	*Bacteroidetes*	*Bacteroidia*	*Bacteroidales*	*Bacteroidaceae*	*Bacteroides*	
88	−2.2583	2.0938 × 10^−2^	1.2943 × 10^−1^	*Bacteria*	*Bacteroidetes*	*Bacteroidia*	*Bacteroidales*	*Bacteroidaceae*	*Bacteroides*	
89	−2.0048	2.1136 × 10^−2^	1.2943 × 10^−1^	*Bacteria*	*Bacteroidetes*	*Bacteroidia*	*Bacteroidales*	*Bacteroidaceae*	*Bacteroides*	
90	1.2044	2.1829 × 10^−2^	1.3187 × 10^−1^	*Bacteria*	*Verrucomicrobia*	*Verrucomicrobiae*	*Verrucomicrobiales*	*Verrucomicrobiaceae*	*Akkermansia*	*muciniphila*
91	−1.9385	2.2019 × 10^−2^	1.3187 × 10^−1^	*Bacteria*	*Firmicutes*	*Clostridia*	*Clostridiales*	*Ruminococcaceae*	*Ruminococcus*	*callidus*
92	1.1277	2.3192 × 10^−2^	1.3739 × 10^−1^	*Bacteria*	*Firmicutes*	*Clostridia*	*Clostridiales*	*Lachnospiraceae*	*Lactonifactor*	*longoviformis*
93	1.5301	2.4958 × 10^−2^	1.4463 × 10^−1^	*Bacteria*	*Firmicutes*	*Bacilli*	*Lactobacillales*	*Lactobacillaceae*	*Lactobacillus*	*delbrueckii*
94	−1.8	2.5154 × 10^−2^	1.4463 × 10^−1^	*Bacteria*	*Firmicutes*	*Clostridia*	*Clostridiales*	*Lachnospiraceae*	*Coprococcus*	
95	1442	2.5211 × 10^−2^	1.4463 × 10^−1^	*Bacteria*	*Firmicutes*	*Clostridia*	*Clostridiales*	*Lachnospiraceae*	*Coprococcus*	
96	−1698	2.8138 × 10^−2^	1.5974 × 10^−1^	*Bacteria*	*Firmicutes*	*Clostridia*	*Clostridiales*	*Lachnospiraceae*	*Blautia*	
97	1.8135	2.8847 × 10^−2^	1.6208 × 10^−1^	*Bacteria*	*Firmicutes*	*Clostridia*	*Clostridiales*	*Ruminococcaceae*	*Butyricicoccus*	*pullicaecorum*
98	2.0172	2.9987 × 10^−2^	1.6676 × 10^−1^	*Bacteria*	*Firmicutes*	*Clostridia*	*Clostridiales*	*Ruminococcaceae*	*Faecalibacterium*	*prausnitzii*
99	1.6882	3.1703 × 10^−2^	1.7367 × 10^−1^	*Bacteria*	*Firmicutes*	*Bacilli*	*Lactobacillales*	*Streptococcaceae*	*Streptococcus*	
100	−1.0322	3.1867 × 10^−2^	1.7367 × 10^−1^	*Bacteria*	*Firmicutes*	*Clostridia*	*Clostridiales*	*Ruminococcaceae*	*Ruminococcus*	
101	−1.0424	3.3676 × 10^−2^	1.8172 × 10^−1^	*Bacteria*	*Bacteroidetes*	*Bacteroidia*	*Bacteroidales*	*Bacteroidaceae*	*Bacteroides*	*plebeius*
102	1.8621	3.5045 × 10^−2^	1.8725 × 10^−1^	*Bacteria*	*Firmicutes*	*Bacilli*	*Lactobacillales*	*Streptococcaceae*	*Streptococcus*	*anginosus*
103	−2.0181	3.7084 × 10^−2^	1.9551 × 10^−1^	*Bacteria*	*Firmicutes*	*Clostridia*	*Clostridiales*	*Lachnospiraceae*	*Blautia*	
104	−1.7395	3.7518 × 10^−2^	1.9551 × 10^−1^	*Bacteria*	*Firmicutes*	*Clostridia*	*Clostridiales*	*Ruminococcaceae*		
105	−1.9389	3.7750 × 10^−2^	1.9551 × 10^−1^	*Bacteria*	*Firmicutes*	*Clostridia*	*Clostridiales*	*Ruminococcaceae*		
106	1.7695	3.8138 × 10^−2^	1.9551 × 10^−1^	*Bacteria*	*Firmicutes*	*Clostridia*	*Clostridiales*	*Veillonellaceae*	*Dialister*	
107	−1.5898	3.8411 × 10^−2^	1.9551 × 10^−1^	*Bacteria*	*Firmicutes*	*Clostridia*	*Clostridiales*	*Lachnospiraceae*		
108	2.0625	3.8743 × 10^−2^	1.9551 × 10^−1^	*Bacteria*	*Firmicutes*	*Bacilli*	*Lactobacillales*	*Lactobacillaceae*	*Lactobacillus*	
109	1.4315	3.9205 × 10^−2^	1.9603 × 10^−1^	*Bacteria*	*Bacteroidetes*	*Bacteroidia*	*Bacteroidales*	*Bacteroidaceae*	*Bacteroides*	
110	−1.0576	4.1726 × 10^−2^	2.0502 × 10^−1^	*Bacteria*	*TM7*	*TM7^−3^*				
111	2.3026	4.1900 × 10^−2^	2.0502 × 10^−1^	*Bacteria*	*Verrucomicrobia*	*Verrucomicrobiae*	*Verrucomicrobiales*	*Verrucomicrobiaceae*	*Akkermansia*	*muciniphila*
112	1.3757	4.2133 × 10^−2^	2.0502 × 10^−1^	*Bacteria*	*Firmicutes*	*Clostridia*	*Clostridiales*	*Veillonellaceae*	*Dialister*	
113	1.7892	4.3243 × 10^−2^	2.0856 × 10^−1^	*Bacteria*	*Firmicutes*	*Clostridia*	*Clostridiales*	*Clostridiaceae*	*Clostridium*	
114	1.3425	4.4328 × 10^−2^	2.1182 × 10^−1^	*Bacteria*	*Firmicutes*	*Clostridia*	*Clostridiales*	*Lachnospiraceae*	*Anaerostipes*	
115	1.6333	4.4696 × 10^−2^	2.1182 × 10^−1^	*Bacteria*	*Firmicutes*	*Clostridia*	*Clostridiales*	*Lachnospiraceae*	*Ruminococcus*	
116	−1.3514	4.5700 × 10^−2^	2.1471 × 10^−1^	*Bacteria*	*Firmicutes*	*Clostridia*	*Clostridiales*	*Lachnospiraceae*		
117	0.95612	4.6789 × 10^−2^	2.1795 × 10^−1^	*Bacteria*	*Bacteroidetes*	*Bacteroidia*	*Bacteroidales*	*Odoribacteraceae*	*Odoribacter*	
118	−1.4848	4.7378 × 10^−2^	2.1882 × 10^−1^	*Bacteria*	*Firmicutes*	*Clostridia*	*Clostridiales*	*Ruminococcaceae*		
119	−1.4123	4.8601 × 10^−2^	2.2258 × 10^−1^	*Bacteria*	*Firmicutes*	*Bacilli*	*Lactobacillales*	*Carnobacteriaceae*	*Granulicatella*	
120	1.0694	4.9042 × 10^−2^	2.2273 × 10^−1^	*Bacteria*	*Firmicutes*	*Clostridia*	*Clostridiales*	*Ruminococcaceae*	*Butyricicoccus*	*pullicaecorum*

**Table 4 nutrients-15-01841-t004:** Changes in the Operational Taxonomic Units (OTUs) induced by the two-phase diet with respect to the baseline and the relative taxonomic classification. log2FC (fold change). FDR: False Discovery Rate.

OTU	log2FC	*p*-Value	FDR	Kingdom	Phylum	Class	Order	Family	Genus	Species
1	−5.4263	8.8618 × 10^−8^	6.3362 × 10^−5^	*Bacteria*	*Verrucomicrobia*	*Verrucomicrobiae*	*Verrucomicrobiales*	*Verrucomicrobiaceae*	*Akkermansia*	*muciniphila*
2	−4.4809	6.8777 × 10^−7^	2.4588 × 10^−4^	*Bacteria*	*Firmicutes*	*Clostridia*	*Clostridiales*	*Ruminococcaceae*	*Gemmiger*	*formicilis*
3	6.3924	2.3738 × 10^−6^	3.2160 × 10^−4^	*Bacteria*	*Verrucomicrobia*	*Verrucomicrobiae*	*Verrucomicrobiales*	*Verrucomicrobiaceae*	*Akkermansia*	*muciniphila*
4	4.0763	2.4129 × 10^−6^	3.2160 × 10^−4^	*Bacteria*	*Bacteroidetes*	*Bacteroidia*	*Bacteroidales*	*Prevotellaceae*	*Prevotella*	*copri*
5	4.4775	3.8963 × 10^−6^	3.2160 × 10^−4^	*Bacteria*	*Bacteroidetes*	*Bacteroidia*	*Bacteroidales*	*Bacteroidaceae*	*Bacteroides*	*fragilis*
6	−3.8154	3.9148 × 10^−6^	3.2160 × 10^−4^	*Bacteria*	*Firmicutes*	*Bacilli*	*Lactobacillales*	*Streptococcaceae*	*Streptococcus*	*luteciae*
7	−4422	3.9707 × 10^−6^	3.2160 × 10^−4^	*Bacteria*	*Proteobacteria*	*Gammaproteobacteria*	*Enterobacteriales*	*Enterobacteriaceae*	*Enterobacter*	
8	4.8582	4.0481 × 10^−6^	3.2160 × 10^−4^	*Bacteria*	*Bacteroidetes*	*Bacteroidia*	*Bacteroidales*	*Bacteroidaceae*	*Bacteroides*	*fragilis*
9	3953	7.4301 × 10^−6^	5.3125 × 10^−4^	*Bacteria*	*Firmicutes*	*Clostridia*	*Clostridiales*	*Lachnospiraceae*	*Lachnobacterium*	
10	3.2155	8.8924 × 10^−6^	5.7801 × 10^−4^	*Bacteria*	*Firmicutes*	*Bacilli*	*Lactobacillales*	*Streptococcaceae*	*Streptococcus*	
11	3.1083	1.7775 × 10^−5^	1.0591 × 10^−3^	*Bacteria*	*Firmicutes*	*Erysipelotrichi*	*Erysipelotrichales*	*Erysipelotrichaceae*	*Bulleidia*	*moorei*
12	3.1819	2.0148 × 10^−5^	1.1081 × 10^−3^	*Bacteria*	*Firmicutes*	*Bacilli*	*Lactobacillales*	*Lactobacillaceae*	*Lactobacillus*	*delbrueckii*
13	3.5205	3.7887 × 10^−5^	1.9349 × 10^−3^	*Bacteria*	*Bacteroidetes*	*Bacteroidia*	*Bacteroidales*	*Porphyromonadaceae*	*Parabacteroides*	*distasonis*
14	−3.6138	6.3042 × 10^−5^	3.0050 × 10^−3^	*Bacteria*	*Bacteroidetes*	*Bacteroidia*	*Bacteroidales*	*Bacteroidaceae*	*Bacteroides*	
15	−4.1915	7.4232 × 10^−5^	3.3172 × 10^−3^	*Bacteria*	*Firmicutes*	*Erysipelotrichi*	*Erysipelotrichales*	*Erysipelotrichaceae*		
16	−4.0131	1.1893 × 10^−4^	4.7241 × 10^−3^	*Bacteria*	*Firmicutes*	*Bacilli*	*Lactobacillales*	*Lactobacillaceae*	*Lactobacillus*	*salivarius*
17	−3.0088	1.7966 × 10^−4^	6.7608 × 10^−3^	*Bacteria*	*Firmicutes*	*Bacilli*	*Lactobacillales*			
18	4.2851	1.9532 × 10^−4^	6.9826 × 10^−3^	*Bacteria*	*Firmicutes*	*Bacilli*	*Lactobacillales*	*Enterococcaceae*	*Enterococcus*	
19	2.9226	2.4910 × 10^−4^	8.4812 × 10^−3^	*Bacteria*	*Firmicutes*	*Clostridia*	*Clostridiales*	*Ruminococcaceae*	*Ruminococcus*	
20	−2.6538	3.2482 × 10^−4^	1.0557 × 10^−2^	*Bacteria*	*Firmicutes*	*Clostridia*	*Clostridiales*	*Ruminococcaceae*	*Ruminococcus*	
21	−2.2606	3.9921 × 10^−4^	1.2410 × 10^−2^	*Bacteria*	*Firmicutes*	*Clostridia*	*Clostridiales*	*Lachnospiraceae*		
22	−2.9928	4.2643 × 10^−4^	1.2704 × 10^−2^	*Bacteria*	*Firmicutes*	*Bacilli*	*Lactobacillales*	*Streptococcaceae*	*Streptococcus*	
23	2.1047	5.2465 × 10^−4^	1.4456 × 10^−2^	*Bacteria*	*Firmicutes*	*Clostridia*	*Clostridiales*	*Veillonellaceae*	*Veillonella*	*parvula*
24	−4.0424	5.2567 × 10^−4^	1.4456 × 10^−2^	*Bacteria*	*Firmicutes*	*Bacilli*	*Lactobacillales*	*Streptococcaceae*	*Streptococcus*	
25	−2.2846	5.5323 × 10^−4^	1.4536 × 10^−2^	*Bacteria*	*Firmicutes*	*Bacilli*	*Lactobacillales*	*Enterococcaceae*	*Enterococcus*	
26	3.0551	5.6923 × 10^−4^	1.4536 × 10^−2^	*Bacteria*	*Firmicutes*	*Clostridia*	*Clostridiales*	*Lachnospiraceae*	*Roseburia*	
27	−2.1389	6.2473 × 10^−4^	1.5403 × 10^−2^	*Bacteria*	*Firmicutes*	*Bacilli*	*Lactobacillales*	*Lactobacillaceae*	*Lactobacillus*	
28	−2.7105	7.8519 × 10^−4^	1.8403 × 10^−2^	*Bacteria*	*Actinobacteria*	*Coriobacteriia*	*Coriobacteriales*	*Coriobacteriaceae*		
29	2.1206	8.1350 × 10^−4^	1.8403 × 10^−2^	*Bacteria*	*Firmicutes*	*Clostridia*	*Clostridiales*	*Ruminococcaceae*	*Ruminococcus*	
30	−2013	8.4910 × 10^−4^	1.8403 × 10^−2^	*Bacteria*	*Firmicutes*	*Clostridia*	*Clostridiales*			
31	−2.3039	8.4936 × 10^−4^	1.8403 × 10^−2^	*Bacteria*	*Firmicutes*	*Clostridia*	*Clostridiales*			
32	1.8956	9.6311 × 10^−4^	1.9003 × 10^−2^	*Bacteria*	*Firmicutes*	*Clostridia*	*Clostridiales*			
33	−1.9534	9.7093 × 10^−4^	1.9003 × 10^−2^	*Bacteria*	*Firmicutes*	*Clostridia*	*Clostridiales*			
34	2.7001	9.8132 × 10^−4^	1.9003 × 10^−2^	*Bacteria*	*Firmicutes*	*Bacilli*	*Lactobacillales*	*Streptococcaceae*	*Lactococcus*	
35	−1.9978	9.8337 × 10^−4^	1.9003 × 10^−2^	*Bacteria*	*Firmicutes*	*Clostridia*	*Clostridiales*	*Ruminococcaceae*	*Butyricicoccus*	*pullicaecorum*
36	−1.7894	1.0170 × 10^−3^	1.9136 × 10^−2^	*Bacteria*	*Firmicutes*	*Clostridia*	*Clostridiales*	*Lachnospiraceae*		
37	3164	1.1950 × 10^−3^	2.1908 × 10^−2^	*Bacteria*	*Firmicutes*	*Clostridia*	*Clostridiales*	*Ruminococcaceae*	*Faecalibacterium*	*prausnitzii*
38	3.4493	1.2854 × 10^−3^	2.2976 × 10^−2^	*Bacteria*	*Firmicutes*	*Clostridia*	*Clostridiales*	*Ruminococcaceae*	*Faecalibacterium*	*prausnitzii*
39	−1.8956	1.5878 × 10^−3^	2.7303 × 10^−2^	*Bacteria*	*Bacteroidetes*	*Bacteroidia*	*Bacteroidales*	*Barnesiellaceae*		
40	−2.3766	1.6038 × 10^−3^	2.7303 × 10^−2^	*Bacteria*	*Firmicutes*	*Clostridia*	*Clostridiales*	*Lachnospiraceae*		
41	−2.2781	2.0958 × 10^−3^	3.4848 × 10^−2^	*Bacteria*	*Firmicutes*	*Clostridia*	*Clostridiales*			
42	−1.7436	2.1690 × 10^−3^	3.5239 × 10^−2^	*Bacteria*	*Firmicutes*	*Clostridia*	*Clostridiales*	*Lachnospiraceae*		
43	2.3781	2.2178 × 10^−3^	3.5239 × 10^−2^	*Bacteria*	*Firmicutes*	*Clostridia*	*Clostridiales*	*Lachnospiraceae*	*Dorea*	
44	−2.3341	2.4462 × 10^−3^	3.7492 × 10^−2^	*Bacteria*	*Firmicutes*	*Clostridia*	*Clostridiales*	*Ruminococcaceae*	*Oscillospira*	
45	−2.7136	2.4645 × 10^−3^	3.7492 × 10^−2^	*Bacteria*	*Firmicutes*	*Clostridia*	*Clostridiales*	*Veillonellaceae*	*Megamonas*	
46	−1.6889	2.7317 × 10^−3^	4.0232 × 10^−2^	*Bacteria*	*Firmicutes*	*Clostridia*	*Clostridiales*	*Ruminococcaceae*	*Oscillospira*	
47	−1542	2.7572 × 10^−3^	4.0232 × 10^−2^	*Bacteria*	*Firmicutes*	*Clostridia*	*Clostridiales*	*Ruminococcaceae*		
48	2.7622	2.8437 × 10^−3^	4.0665 × 10^−2^	*Bacteria*	*Firmicutes*	*Clostridia*	*Clostridiales*	*Veillonellaceae*	*Dialister*	
49	−2.7669	2.9114 × 10^−3^	4.0817 × 10^−2^	*Bacteria*	*Firmicutes*	*Clostridia*	*Clostridiales*	*Ruminococcaceae*	*Gemmiger*	*formicilis*
50	3.6484	3.0310 × 10^−3^	3.2160 × 10^−4^	*Bacteria*	*Bacteroidetes*	*Bacteroidia*	*Bacteroidales*	*Bacteroidaceae*	*Bacteroides*	*ovatus*
51	1.6014	3.0865 × 10^−3^	4.0836 × 10^−2^	*Bacteria*	*Firmicutes*	*Clostridia*	*Clostridiales*	*Lachnospiraceae*		
52	1.8548	3.1060 × 10^−3^	4.0836 × 10^−2^	*Bacteria*	*Bacteroidetes*	*Bacteroidia*	*Bacteroidales*	*Paraprevotellaceae*	*Paraprevotella*	
53	−1.6527	3.1159 × 10^−3^	4.0836 × 10^−2^	*Bacteria*	*Firmicutes*	*Clostridia*	*Clostridiales*	*Lachnospiraceae*		
54	1.6715	3.1412 × 10^−3^	4.0836 × 10^−2^	*Bacteria*	*Verrucomicrobia*	*Verrucomicrobiae*	*Verrucomicrobiales*	*Verrucomicrobiaceae*	*Akkermansia*	*muciniphila*
55	1.4681	3.5714 × 10^−3^	4.5600 × 10^−2^	*Bacteria*	*Bacteroidetes*	*Bacteroidia*	*Bacteroidales*	*S24^−7^*		
56	1.8228	3.6913 × 10^−3^	4.6303 × 10^−2^	*Bacteria*	*Firmicutes*	*Clostridia*	*Clostridiales*	*Ruminococcaceae*	*Oscillospira*	
57	2.1704	3.9970 × 10^−3^	4.9274 × 10^−2^	*Bacteria*	*Bacteroidetes*	*Bacteroidia*	*Bacteroidales*	*Porphyromonadaceae*	*Parabacteroides*	*distasonis*
58	−1.8689	4.3246 × 10^−3^	5.2409 × 10^−2^	*Bacteria*	*Bacteroidetes*	*Bacteroidia*	*Bacteroidales*	*Bacteroidaceae*	*Bacteroides*	*caccae*
59	1.9413	4.5257 × 10^−3^	5.3931 × 10^−2^	*Bacteria*	*Firmicutes*	*Clostridia*	*Clostridiales*	*Lachnospiraceae*		
60	−1491	5.5162 × 10^−3^	6.3748 × 10^−2^	*Bacteria*	*Firmicutes*	*Clostridia*	*Clostridiales*	*Lachnospiraceae*	*Coprococcus*	
61	−2504	5.5278 × 10^−3^	6.3748 × 10^−2^	*Bacteria*	*Bacteroidetes*	*Bacteroidia*	*Bacteroidales*	*Porphyromonadaceae*	*Parabacteroides*	*distasonis*
62	2.2676	6.2165 × 10^−3^	7.0552 × 10^−2^	*Bacteria*	*Bacteroidetes*	*Bacteroidia*	*Bacteroidales*	*Paraprevotellaceae*	*Prevotella*	
63	−2.3658	6.5409 × 10^−3^	7.3075 × 10^−2^	*Bacteria*	*Firmicutes*	*Clostridia*	*Clostridiales*	*Veillonellaceae*	*Megamonas*	
64	−1472	6.9404 × 10^−3^	7.5887 × 10^−2^	*Bacteria*	*Firmicutes*	*Clostridia*	*Clostridiales*	*Ruminococcaceae*	*Oscillospira*	
65	−2.4861	7.0050 × 10^−3^	7.5887 × 10^−2^	*Bacteria*	*Actinobacteria*	*Actinobacteria*	*Bifidobacteriales*	*Bifidobacteriaceae*	*Bifidobacterium*	*animalis*
66	1.3615	7.5573 × 10^−3^	8.0648 × 10^−2^	*Bacteria*	*Bacteroidetes*	*Bacteroidia*	*Bacteroidales*	*Rikenellaceae*	*Alistipes*	*finegoldii*
67	−2.2229	7.8907 × 10^−3^	8.2968 × 10^−2^	*Bacteria*	*Firmicutes*	*Clostridia*	*Clostridiales*	*Lachnospiraceae*		
68	1.3963	9.3224 × 10^−3^	9.6601 × 10^−2^	*Bacteria*	*Firmicutes*	*Clostridia*	*Clostridiales*	*Lachnospiraceae*		
69	−1.6429	9.5803 × 10^−3^	9.7856 × 10^−2^	*Bacteria*	*Proteobacteria*	*Betaproteobacteria*	*Burkholderiales*	*Alcaligenaceae*	*Sutterella*	
70	2.4653	9.9878 × 10^−3^	9.9350 × 10^−2^	*Bacteria*	*Proteobacteria*	*Gammaproteobacteria*	*Pasteurellales*	*Pasteurellaceae*	*Haemophilus*	*parainfluenzae*
71	−2.2795	1.0005 × 10^−2^	9.9350 × 10^−2^	*Bacteria*	*Firmicutes*	*Clostridia*	*Clostridiales*			
72	2132	1.0350 × 10^−2^	1.0137 × 10^−1^	*Bacteria*	*Firmicutes*	*Clostridia*	*Clostridiales*	*Ruminococcaceae*	*Faecalibacterium*	*prausnitzii*
73	−2.1206	1.0607 × 10^−2^	1.0249 × 10^−1^	*Bacteria*	*Firmicutes*	*Clostridia*	*Clostridiales*			
74	−1.9426	1.1121 × 10^−2^	1.0486 × 10^−1^	*Bacteria*	*Actinobacteria*	*Coriobacteriia*	*Coriobacteriales*	*Coriobacteriaceae*		
75	−1.3202	1.1146 × 10^−2^	1.0486 × 10^−1^	*Bacteria*	*Firmicutes*	*Clostridia*	*Clostridiales*	*Lachnospiraceae*	*Coprococcus*	
76	2.0855	1.1590 × 10^−2^	1.0695 × 10^−1^	*Bacteria*	*Firmicutes*	*Clostridia*	*Clostridiales*	*Lachnospiraceae*	*Blautia*	*producta*
77	−1297	1.1668 × 10^−2^	1.0695 × 10^−1^	*Bacteria*	*Firmicutes*	*Bacilli*	*Lactobacillales*	*Streptococcaceae*	*Streptococcus*	
78	−1.9909	1.2041 × 10^−2^	1.0898 × 10^−1^	*Bacteria*	*Proteobacteria*	*Gammaproteobacteria*	*Enterobacteriales*	*Enterobacteriaceae*	*Escherichia*	*coli*
79	1.4621	1.2325 × 10^−2^	1.1015 × 10^−1^	*Bacteria*	*Firmicutes*	*Erysipelotrichi*	*Erysipelotrichales*	*Erysipelotrichaceae*	*Clostridium*	*spiroforme*
80	−1.53	1.2565 × 10^−2^	1.1091 × 10^−1^	*Bacteria*	*Firmicutes*	*Clostridia*	*Clostridiales*	*Ruminococcaceae*	*Oscillospira*	
81	1.8255	1.2976 × 10^−2^	1.1314 × 10^−1^	*Bacteria*	*Firmicutes*	*Clostridia*	*Clostridiales*	*Lachnospiraceae*	*Ruminococcus*	
82	−2.0635	1.3867 × 10^−2^	1.1946 × 10^−1^	*Bacteria*	*Firmicutes*	*Erysipelotrichi*	*Erysipelotrichales*	*Erysipelotrichaceae*	*Eubacterium*	*cylindroides*
83	−1.7321	1.5107 × 10^−2^	1.2859 × 10^−1^	*Bacteria*	*Firmicutes*	*Clostridia*	*Clostridiales*	*Ruminococcaceae*		
84	−1.1892	1.7496 × 10^−2^	1.4454 × 10^−1^	*Bacteria*	*Firmicutes*	*Clostridia*	*Clostridiales*	*Ruminococcaceae*	*Ruminococcus*	
85	1.9963	1.7668 × 10^−2^	1.4454 × 10^−1^	*Bacteria*	*Firmicutes*	*Clostridia*	*Clostridiales*	*Christensenellaceae*		
86	−1.8295	1.7773 × 10^−2^	1.4454 × 10^−1^	*Bacteria*	*Firmicutes*	*Clostridia*	*Clostridiales*	*Lachnospiraceae*		
87	−1235	1.7790 × 10^−2^	1.4454 × 10^−1^	*Bacteria*	*Firmicutes*	*Erysipelotrichi*	*Erysipelotrichales*	*Erysipelotrichaceae*	*Eubacterium*	
88	−2.1025	1.8242 × 10^−2^	1.4655 × 10^−1^	*Bacteria*	*Firmicutes*	*Clostridia*	*Clostridiales*	*Lachnospiraceae*		
89	−1.4924	1.8574 × 10^−2^	1.4741 × 10^−1^	*Bacteria*	*Bacteroidetes*	*Bacteroidia*	*Bacteroidales*	*Bacteroidaceae*	*Bacteroides*	*plebeius*
90	−1.1541	1.8762 × 10^−2^	1.4741 × 10^−1^	*Bacteria*	*Firmicutes*	*Clostridia*	*Clostridiales*			
91	−1.8936	1.9680 × 10^−2^	1.5169 × 10^−1^	*Bacteria*	*Firmicutes*	*Clostridia*	*Clostridiales*	*Lachnospiraceae*	*Roseburia*	*faecis*
92	−2.2873	1.9730 × 10^−2^	1.5169 × 10^−1^	*Bacteria*	*Firmicutes*	*Erysipelotrichi*	*Erysipelotrichales*	*Erysipelotrichaceae*	*Eubacterium*	*cylindroides*
93	−2.3029	2.0475 × 10^−2^	1.5574 × 10^−1^	*Bacteria*	*Firmicutes*	*Erysipelotrichi*	*Erysipelotrichales*	*Erysipelotrichaceae*	*Catenibacterium*	
94	−1.0769	2.1866 × 10^−2^	1.6457 × 10^−1^	*Bacteria*	*Firmicutes*	*Erysipelotrichi*	*Erysipelotrichales*	*Erysipelotrichaceae*	*cc_115*	
95	−1.2226	2.3023 × 10^−2^	1.7147 × 10^−1^	*Bacteria*	*Firmicutes*	*Clostridia*	*Clostridiales*	*Tissierellaceae*	*WAL_1855D*	
96	−1.8686	2.3717 × 10^−2^	1.7482 × 10^−1^	*Bacteria*	*Firmicutes*	*Bacilli*	*Turicibacterales*	*Turicibacteraceae*	*Turicibacter*	
97	−1.2382	2.5469 × 10^−2^	1.8582 × 10^−1^	*Bacteria*	*Bacteroidetes*	*Bacteroidia*	*Bacteroidales*	*Paraprevotellaceae*		
98	−1.07	3.0049 × 10^−2^	2.1702 × 10^−1^	*Bacteria*	*Firmicutes*	*Clostridia*	*Clostridiales*	*Ruminococcaceae*	*Ruminococcus*	
99	−1.2621	3.1443 × 10^−2^	2.2283 × 10^−1^	*Bacteria*	*Firmicutes*	*Clostridia*	*Clostridiales*	*Mogibacteriaceae*		
100	−1.0366	3.1476 × 10^−2^	2.2283 × 10^−1^	*Bacteria*	*Firmicutes*	*Clostridia*	*Clostridiales*	*Ruminococcaceae*		
101	−1.2642	3.2767 × 10^−2^	2.2770 × 10^−1^	*Bacteria*	*Firmicutes*	*Clostridia*	*Clostridiales*	*Lachnospiraceae*	*Ruminococcus*	
102	1.0243	3.2801 × 10^−2^	2.2770 × 10^−1^	*Bacteria*	*Firmicutes*	*Bacilli*	*Lactobacillales*	*Streptococcaceae*	*Lactococcus*	*garvieae*
103	1.2852	3.3817 × 10^−2^	2.3249 × 10^−1^	*Bacteria*	*Firmicutes*	*Clostridia*	*Clostridiales*	*Ruminococcaceae*		
104	1.6256	3.4380 × 10^−2^	2.3411 × 10^−1^	*Bacteria*	*Firmicutes*	*Clostridia*	*Clostridiales*			
105	1.3859	3.4876 × 10^−2^	2.3525 × 10^−1^	*Bacteria*	*Bacteroidetes*	*Bacteroidia*	*Bacteroidales*	*Bacteroidaceae*	*Bacteroides*	*eggerthii*
106	−1.3103	3.6169 × 10^−2^	2.4034 × 10^−1^	*Bacteria*	*Firmicutes*	*Clostridia*	*Clostridiales*	*Clostridiaceae*	*Clostridium*	*perfringens*
107	−1.7195	3.6303 × 10^−2^	2.4034 × 10^−1^	*Bacteria*	*Firmicutes*	*Clostridia*	*Clostridiales*	*Lachnospiraceae*	*Coprococcus*	
108	0.99229	3.6895 × 10^−2^	2.4202 × 10^−1^	*Bacteria*	*Firmicutes*	*Clostridia*	*Clostridiales*	*Ruminococcaceae*		
109	−1.7076	3.7308 × 10^−2^	2.4250 × 10^−1^	*Bacteria*	*Bacteroidetes*	*Bacteroidia*	*Bacteroidales*	*Porphyromonadaceae*	*Parabacteroides*	
110	1.6698	3.7650 × 10^−2^	2.4252 × 10^−1^	*Bacteria*	*Bacteroidetes*	*Bacteroidia*	*Bacteroidales*	*Bacteroidaceae*	*Bacteroides*	
111	1.0782	3.8197 × 10^−2^	2.4385 × 10^−1^	*Bacteria*	*Proteobacteria*	*Alphaproteobacteria*	*RF32*			
112	−1.9231	4.1572 × 10^−2^	2.5858 × 10^−1^	*Bacteria*	*Bacteroidetes*	*Bacteroidia*	*Bacteroidales*	*Bacteroidaceae*	*Bacteroides*	*ovatus*
113	−1.4079	4.1652 × 10^−2^	2.5858 × 10^−1^	*Bacteria*	*Bacteroidetes*	*Bacteroidia*	*Bacteroidales*	*Rikenellaceae*	*Alistipes*	*onderdonkii*
114	−1.1447	4.1727 × 10^−2^	2.5858 × 10^−1^	*Bacteria*	*Firmicutes*	*Clostridia*	*Clostridiales*	*Lachnospiraceae*		
115	1.6189	4.1951 × 10^−2^	2.5858 × 10^−1^	*Bacteria*	*Firmicutes*	*Clostridia*	*Clostridiales*	*Ruminococcaceae*	*Faecalibacterium*	*prausnitzii*
116	1.1002	4.2355 × 10^−2^	2.5883 × 10^−1^	*Bacteria*	*Firmicutes*	*Clostridia*	*Clostridiales*	*Lachnospiraceae*		
117	0.95781	4.3220 × 10^−2^	2.6188 × 10^−1^	*Bacteria*	*Actinobacteria*	*Actinobacteria*	*Actinomycetales*	*Corynebacteriaceae*	*Corynebacterium*	*variabile*
118	1.2058	4.6079 × 10^−2^	2.7686 × 10^−1^	*Bacteria*	*Firmicutes*	*Clostridia*	*Clostridiales*	*Lachnospiraceae*		
119	−0.947	4.6507 × 10^−2^	2.7711 × 10^−1^	*Bacteria*	*Firmicutes*	*Clostridia*	*Clostridiales*	*Christensenellaceae*		
120	1.6527	4.7378 × 10^−2^	2.7798 × 10^−1^	*Bacteria*	*Bacteroidetes*	*Bacteroidia*	*Bacteroidales*	*Barnesiellaceae*		
121	1.6087	4.7731 × 10^−2^	2.7798 × 10^−1^	*Bacteria*	*Firmicutes*	*Clostridia*	*Clostridiales*	*Lachnospiraceae*		
122	1.0063	4.7821 × 10^−2^	2.7798 × 10^−1^	*Bacteria*	*Firmicutes*	*Clostridia*	*Clostridiales*	*Ruminococcaceae*	*Anaerotruncus*	

## Data Availability

Research data are available upon request by contacting the corresponding author of the article.

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
