# Peer review of "Modulation of Gut Microbiota through Low-Calorie and Two-Phase Diets in Obese Individuals"

_nutrients, 2023, doi:10.3390/nu15081841_

Round 1
Reviewer 1 Report
This paper used two kinds of diet intervention and did 16sRNA sequencing and analysis to check the changes of gut microbiota in obesity individuals. However , this topic is lack of novelty and there's no rational to compare two different regimens of nutrition intervention. The design of the experiments has a lot of problems. For the low calorie diet, they should have provide the nutrition component of the diet and if the food had been changed, they should provide paired food for a paired group to clarify that it is not food components that changed gut biota. Second, for the two- phase diet intervention, they should give some references for why using this kind of change and make another pair of food that start with low- calorie diet and followed by ketogenic food. The conclusion of this paper is too simple and data is not solid so they can not explain clearly how nutritional intervention change affect microbiota.
Author Response
This paper used two kinds of diet intervention and did 16sRNA sequencing and analysis to check the changes of gut microbiota in obesity individuals. However, this topic is lack of novelty and there's no rational to compare two different regimens of nutrition intervention. The design of the experiments has a lot of problems. For the low calorie diet, they should have provide the nutrition component of the diet and if the food had been changed, they should provide paired food for a paired group to clarify that it is not food components that changed gut biota. Second, for the two- phase diet intervention, they should give some references for why using this kind of change and make another pair of food that start with low- calorie diet and followed by ketogenic food. The conclusion of this paper is too simple and data is not solid so they can not explain clearly how nutritional intervention change affect microbiota.
Response: we thank the reviewer for the comments, and we agree on the need to better clarify the rationale of the study and detail the components of the two diets. As a result, the manuscript has been thoroughly revised to enrich it and help readability. In paragraph 2.1 of Materials and Methods of the revised version of the manuscript, we have introduced Table S1, which reports in detail the daily meal plan of the two diets. What is more, in the same paragraph, we have highlighted that, in neither of the two diets, we did vary the foods to the subjects but in both we adopted a balanced nutritional plan that reduced the caloric intake by preserving components that provide adequate amounts of carbohydrates, lipids, proteins, minerals, and vitamins.
As for the two phase-diet, we did not report references concerning this nutritional approach as, although conceptually similar, the experimental conditions reported in the literature are different from each other and ours. However, we share the reviewer's observation so we have introduced some of these in the paragraph “Discussion” of the revised version of the manuscript.
In the paragraphs “Discussion” and “Conclusions”, we addressed all reviewer's comments.
Reviewer 2 Report
Thanks for all the hard work. Very interesting article. Good luck

Author Response
Thanks for all the hard work. Very interesting article. Good luck
Response: we thank the reviewer for the comments
Reviewer 3 Report
Carelli et al. present an interesting study with a high significance for clinical practice, namely the modulation of gut microbiota in obese individuals through two types of diet. However, several questions and comments should be addressed before considering the paper for proceed.
The composition of gut microbiota of non-obese individuals should be included as a control.
The age of the participants is not mentioned. A table or graph summarizing the anthropometric measurements should be presented in order better readability and understanding of the data.
The data of blood biochemical measurements are not presented nor discussed. It will be interesting if some changes are observed, incl. inflammatory markers.
Such presented data (figure 4 and 5) seems that the baselines in the two patients’ groups are different. The authors should present a figure in which the baseline (which should be similar in obese patients before diets) is presented and compared with the two diet types.
A rationale for choosing a two-step diet should be given.
It will be very valuable if the composition of the two diets is described in more detail. Authors could include a specific composition of the menu.
The authors should discuss if and how the intake of probiotics affects the composition of the gut microbiota.
Authors should discuss the potential biochemical mechanisms of how Proteobacteria is involved in obesity. A figure, visualizing the mechanism will be very beneficial.
Author Response
Carelli et al. present an interesting study with a high significance for clinical practice, namely the modulation of gut microbiota in obese individuals through two types of diet. However, several questions and comments should be addressed before considering the paper for proceed.
- The composition of gut microbiota of non-obese individuals should be included as a control.
Response: the reviewer is right and, perhaps, the absence of results relating to non-obese samples could be a limitation of this work. In this regard, however, we want to specify that it was not our purpose to disentangle the differences in microbiota composition between obese and non-obese subjects, which have been widely described in the literature, but rather to investigate the effects of two different nutritional approaches on the gut microbiota in obesity. In any case, following the suggestion of the reviewer, in the paragraph “Discussion” of the revised version of the manuscript, we have commented on the main differences that characterize the gut microbial communities in healthy and obese subjects based on literature data.
- The age of the participants is not mentioned. A table or graph summarizing the anthropometric measurements should be presented in order better readability and understanding of the data.
Response: we thank the reviewer for the suggestions. We have inserted in the paragraph “Results” of the revised version of the manuscript, two tables (Table 1 and Table 2) reporting mean values, standard deviations, and p values of anthropometric and clinical data, and two supplementary tables (Table S2 and Table S3) showing anthropometric and clinical measurements of the study participants.
In paragraph 2.1 of “Materials and Methods” of the revised version of the manuscript, we added the mean age of the participants.
- The data of blood biochemical measurements are not presented nor discussed. It will be interesting if some changes are observed, incl. inflammatory markers.
Response: we thank the reviewer for the observation. In the paragraph “Results” of the revised version of the manuscript, we have inserted Table S3 in which all clinical measurements, that we had available, were listed and we have commented on them accordingly.
- Such presented data (figure 4 and 5) seems that the baselines in the two patients’ groups are different. The authors should present a figure in which the baseline (which should be similar in obese patients before diets) is presented and compared with the two diet types.
- Response: we thank the reviewer for the observation. We have replaced, in the revised version of the manuscript, the two Figures 4 and 5 with a single Figure 4, in which the baseline was presented and compared with the two diets.
- A rationale for choosing a two-step diet should be given.
Response: we thank the reviewer for the comment. We realized that the rationale for choosing the two diets was not well reported in the paper. Therefore, we have commented on it in the paragraph “Discussion” of the revised version of the manuscript.
- It will be very valuable if the composition of the two diets is described in more detail. Authors could include a specific composition of the menu.
Response: we thank the reviewer for the observation. In the paragraph “Material and Methods” of the revised version of the manuscript, we detailed the composition of the two diets and we inserted Table S1 which reports the daily meal plan of the two diets.
- The authors should discuss if and how the intake of probiotics affects the composition of the gut microbiota.
Response: we thank the reviewer for the comment. In the paragraph “Discussion” of the revised version of the manuscript, we commented on the role exerted by probiotics on gut microbiota.
- Authors should discuss the potential biochemical mechanisms of how Proteobacteria is involved in obesity. A figure, visualizing the mechanism will be very beneficial.
Response: we thank the reviewer for the comment. In the paragraph “Discussion” of the revised version of the manuscript, we commented on the role of Proteobacteria in obesity.
Reviewer 4 Report
In this paper, the authors evaluated the effects of two nutritional interventions in modulating the gut microbiota in a population sample of obese subjects.
The results demonstrate that the adoption of specific nutritional interventions associated with the administration of effective probiotics may modify the structure of the gut microbiota, affecting bacteria which functions have been demonstrated to be correlated with the health status in humans. The topic is interesting and the paper is well prepared. There are some concerns that need to be considered.
1. Tables should be three-line tables.
2. The font size in some figures (especially Figure) is too small. Make them larger for easier reading.
3. Line 65, “SCFAs” instead of “SCFA”.
4. Line 68, what’s the full name of “ROS”?
5. Line 148, it’s the first occurrence of the abbreviation of “OTU”.
6. The quantity of references may be too large (74) for a research article.
Author Response
In this paper, the authors evaluated the effects of two nutritional interventions in modulating the gut microbiota in a population sample of obese subjects.
The results demonstrate that the adoption of specific nutritional interventions associated with the administration of effective probiotics may modify the structure of the gut microbiota, affecting bacteria which functions have been demonstrated to be correlated with the health status in humans. The topic is interesting and the paper is well prepared. There are some concerns that need to be considered.
- Tables should be three-line tables.
- The font size in some figures (especially Figure) is too small. Make them larger for easier reading.
- Line 65, “SCFAs” instead of “SCFA”.
- Line 68, what’s the full name of “ROS”?
- Line 148, it’s the first occurrence of the abbreviation of “OTU”.
- The quantity of references may be too large (74) for a research article.
Response: we thank the reviewer. All the reviewer's concerns have been considered in the revised version of the manuscript.
Round 2
Reviewer 1 Report
After this revision I think this paper is suitable to be published in the journal.
Author Response
1. Lines 197-190: a non-significant change means no change and so cannot be described as a "slight reduction"; please adjust. Comment also applies to lines 237-240. Please check throughout manuscript.
2. Lines 202-206: OTUs are integers and so cannot be measured to 3 decimal places. Please adjust.
Response: thanks to the reviewers for the comments. In the revised version of the manuscript, we addressed all reviewer's comments.
Reviewer 3 Report
-
Author Response
- Lines 197-190: a non-significant change means no change and so cannot be described as a "slight reduction"; please adjust. Comment also applies to lines 237-240. Please check throughout manuscript.
- Lines 202-206: OTUs are integers and so cannot be measured to 3 decimal places. Please adjust.
Response: thanks to the reviewers for the comments. In the revised version of the manuscript, we addressed all reviewer's comments.